# Comprehensive Metabolomics in Mouse Mast Cell Model of Allergic Rhinitis for Profiling, Modulation, Semiquantitative Analysis, and Pathway Analysis

**DOI:** 10.3390/biom15010109

**Published:** 2025-01-11

**Authors:** Akshay Suresh Patil, Yan Xu

**Affiliations:** Department of Chemistry, Cleveland State University, Cleveland, OH 44115, USA; a.patil21@vikes.csuohio.edu

**Keywords:** UHPLC-QTOF-MS-based untargeted and targeted metabolomics, semiquantitative analysis, mast cell modulation, pathway analysis, therapeutic agent testing

## Abstract

Allergic rhinitis affects millions globally, causing significant discomfort and reducing the quality of life. This study investigates the metabolic alterations in murine mast cells (MC/9) under allergic rhinitis conditions induced by lipopolysaccharide (LPS) stimulation, employing UHPLC-QTOF-MS-based untargeted and targeted metabolomics. The analysis identified 44 significantly regulated metabolites, including histamine, leukotrienes, prostaglandins, thromboxanes, and ceramides. Key metabolic pathways such as arachidonic acid, histidine, and sphingolipid metabolisms were notably modulated. The study further examined the therapeutic effects of triprolidine and zileuton, demonstrating their capacity to reverse LPS-induced metabolic shifts. Triprolidine primarily modulated histidine and sphingolipid metabolism, while zileuton targeted arachidonic acid and sphingolipid metabolism. These findings underscore the utility of metabolomics analysis in elucidating the complex biochemical pathways involved in allergic rhinitis and highlight the potential of metabolomics for evaluating therapeutic interventions. This study enhances our understanding of mast cell metabolism in allergic responses and provides a robust model for assessing the efficacy of anti-allergic agents, paving the way for more effective treatments.

## 1. Introduction

Allergic rhinitis, a prevalent allergic disorder, affects approximately 80 million Americans each year, manifesting through symptoms such as sneezing, itching, nasal congestion, and rhinorrhea. These symptoms result from inflammation in the nasal passages triggered by common allergens like pollen, hay, and dust [1]. The condition not only alters the body’s immune response, leading to dizziness, fatigue, and body pain [2], but also significantly impacts the quality of life.

Mast cells, the key effectors in the immune system, play a pivotal role in the pathophysiology of allergic rhinitis. These cells release various mediators upon activation, contributing to the characteristic symptoms of allergic rhinitis. As primary sources of cytokines, chemokines, and lipid mediators, mast cells significantly influence immune regulation and inflammation [3]. Deciphering the specific mediators released from mast cells is pivotal in identifying targets and pathways for effective therapeutic strategies.

Certain drugs are designed to target specific pathways within the immune response to manage allergic and inflammatory conditions. Triprolidine and zileuton are two such drugs, each with a distinct mechanism of action. Triprolidine, an antihistamine, blocks histamine H1 receptors [4], reducing symptoms like itching and swelling. Zileuton, a leukotriene synthesis inhibitor, targets 5-lipoxygenase to prevent leukotriene production [5,6], which is crucial in asthma and allergic reactions. In LPS-stimulated mast cells, these drugs exhibit a complementary effect: triprolidine reduces histamine release to mitigate the immediate allergic response, whereas zileuton decreases the production of pro-inflammatory leukotrienes, potentially reducing longer-term inflammation. Therefore, triprolidine and zileuton were employed as positive controls to monitor the metabolomic alterations under allergic rhinitis conditions.

Metabolomics, the study of metabolites—small-molecule substrates, intermediates, and products of metabolism in cells, tissues, or organisms—offers insights into the downstream effects of genomic and proteomic changes. As metabolites are the end products of cellular processes, metabolomics provides a snapshot of the cell’s physiological state, emerging as a critical tool in biomedical and clinical research [7]. This field encompasses both targeted and untargeted approaches. Targeted metabolomics is dedicated to quantifying specific metabolites from a predefined set, whereas untargeted metabolomics provides a broader view, analyzing a wide spectrum of metabolites without pre-existing knowledge about their identities. Consequently, untargeted metabolomics is particularly advantageous for exploring unknown biomarkers and metabolic pathways, thereby unraveling the complexities of allergic rhinitis at the molecular level [8].

The advancements in metabolomics methodologies, notably the adoption of UHPLC-QTOF-MS/MS systems, have substantially improved metabolite identification’s sensitivity, resolution, and speed. These high-resolution techniques enable accurate mass measurements and fragmentation patterns, facilitating precise metabolite identification [9,10]. Traditional methods like ELISA and Western blot, while useful in studying specific proteins associated with allergic responses, lack the capacity to provide a global view of cellular metabolism [11,12]. In contrast, UHPLC-QTOF-MS/MS-based metabolomics allows for the simultaneous detection of a wide range of metabolites, offering a more comprehensive understanding of the molecular events in mast cells during allergic rhinitis. This method complements traditional protein-centric approaches, adding a crucial layer of information for a complete understanding of cellular responses in allergic conditions and enabling the investigation of pathways and semiquantitative analysis of metabolites.

This study focuses on developing an allergic rhinitis model using cell metabolomics. In contrast to previous mast cell studies that primarily employed targeted metabolomics approaches, such as ELISA, SeaHorse, and other techniques [13,14,15], we utilize UHPLC-QTOF-MS/MS technology for comprehensive metabolite profiling, identification, and quantification. This advanced approach allows us to measure a broad spectrum of metabolites under various experimental conditions, providing insight into the metabolic pathways affected by allergy induction and therapeutic interventions. Notably, this is the first report utilizing UHPLC-QTOF-MS-based untargeted metabolomics in a mast cell model of allergic rhinitis. This model can be a valuable tool for developing and evaluating therapeutic agents for allergic rhinitis.

## 2. Materials and Methods

### 2.1. Chemical and Biological Reagents

LC-MS-grade acetonitrile, methanol, and 99.8% pure acetic acid from Acros Organics were sourced from Thermo Fisher Scientific (Waltham, MA, USA). Deionized water was prepared using a Barnstead GenPure xCAD ultrapure water purification system from Thermo Fisher Scientific, ensuring a resistance of 18.2 MΩ. Dimethyl sulfoxide (DMSO) (BioUltra for molecular biology) with a purity exceeding 99.5%, along with ammonium acetate, triprolidine hydrochloride (Product # T6764), and zileuton (Product # PHR 2555), were procured from Sigma Aldrich (St. Louis, MO, USA). Prostaglandin E2-d4 (PGE2-d4) standard solution (500 μg/mL, in methyl acetate) with a purity above 99% was purchased from Cayman Chemical (Ann Arbor, MI, USA) and used as the internal standard (IS) master stock for this work.

Murine mast cells (MC/9) were purchased from the American Type Culture Collection (ATCC) (Manassas, VA, USA). We acquired Gibco™ L-glutamine (200 mM, 100X) and Cosmic calf™ serum from Thermo Fisher Scientific, and Dulbecco’s Modified Eagle Medium (DMEM), 2-mercaptoethanol (BioReagent, purity 99%), and lipopolysaccharide (LPS) (*Escherichia coli O55:B5*) from Sigma Aldrich. Additionally, we used D-PBS (1X) obtained from the Cleveland Clinic (Cleveland, OH, USA).

The working stock solution of PGE2-d4 was prepared at a concentration of 10.0 μg/mL. This was achieved by diluting 20.0 μL of the 500 μg/mL master stock solution with 980 μL of a mixed solvent (acetonitrile/methanol/water in a 2:2:1 ratio). The master stock solutions of triprolidine (1.00 mM) and zileuton (1.00 mM) were prepared by dissolving the respective compounds accurately in 1.00 mL of sterile water and DMSO, respectively. All solutions were stored in the dark at −80 °C when not used.

### 2.2. Cell Culture and Studies

MC/9 cells were cultured in 10.0 cm^3^ tissue culture plates (VWR, Radnor, PA, USA). Each plate contained 10.0 mL of cell culture medium, formulated with 10% Cosmic calf^®^ serum, 2 mM L-glutamine, and 0.05 mM 2-mercaptoethanol. The cultures were maintained in a humidified incubator at 37 °C with a 5% CO_2_ atmosphere, and the medium was renewed every three days.

Our experimental design encompassed four conditions: control, stimulated, and two treatment groups with positive controls (triprolidine and zileuton). The cells were counted using a hemocytometer and microscope with trypan blue staining solution (0.4%). Two biological replicates were used for each experimental condition. The exact number of cells (5 × 10^6^) was used for every biological replicate for each experimental condition. The control group cells were maintained solely in the culture medium without additives for 20 h. For the stimulated group, allergic rhinitis conditions were induced using LPS at a final concentration of 1.00 μg/mL in the culture plates for 4 h. Triprolidine and zileuton, used as positive controls, were applied to the stimulated cells at the final concentrations of 40.0 μM and 0.50 μM, respectively, for 12 h, following the protocols established in previous studies [16,17].

The cells from each experimental condition were collected and transferred to 15.0 mL centrifuge tubes from VWR. A centrifugation step was performed at 1000× *g* for 5 min to separate the cells from the medium. The cell pellets were washed with ice-cold PBS (1×, pH 7.4) and rinsed with ice-cold water to remove residual PBS. Finally, the washed cell pellets were stored at −80 °C until further analysis.

### 2.3. Cell Sample Preparation

Each cell pellet was suspended in 1.00 mL of ice-cold deionized water in glass culture tubes. These glass culture tubes were submerged in a beaker containing ice and sonicated for 10 s (2s/cycle × 5 cycles) using a sonifier from Branson Ultrasonics (Danbury, CT, USA) to lyse the cells by disrupting the cell membrane and denature proteins. The protein concentration in the cell lysate was measured using a Pierce™ BCA protein assay kit from Thermo Fisher Scientific following the manufacturer’s instructions to normalize the cell growth factors across the samples. The final protein concentration in each cell lysate was adjusted to 100 μg/mL by adding deionized water.

A protein precipitation procedure was employed for metabolite extraction. A total of 1.00 mL of each cell lysate was mixed with 2 mL of ice-cold acetonitrile and 1 mL of ice-cold methanol. The sample mixture was vigorously vortexed for 5 min, then stored at −20 °C overnight to maximize protein precipitation. The following day, the samples were centrifuged at 13,000× *g* for 5 min at 4 °C using a Sorvall Legend XTR centrifuge from Thermo Fisher Scientific and the supernatants were collected in fresh borosilicate glass culture tubes (16 × 100 mm) from Thermo Fisher Scientific (Pittsburgh, PA, USA). The samples were dried in an ice bath using an N-EVAP^TM^ 111 nitrogen evaporator from Organomation (West Berlin, MA, USA). Once dried, each sample was reconstituted in 120 μL of a reconstitution solvent (acetonitrile/methanol/water in a 2:2:1 ratio), and 30.0 μL of PGE2-d4 working stock solution was added to achieve a final IS concentration of 2.00 μg/mL. The prepared samples were then transferred into HPLC vials for the subsequent untargeted metabolomic analysis.

### 2.4. UHPLC-QTOF-MS/MS System

This study employed an Agilent 1290 Infinity II UHPLC system coupled with an Agilent 6545 QTOF mass spectrometer (Santa Clara, CA, USA). The UHPLC setup included essential components such as a solvent reservoir, degasser, binary pump, multisampler, and a column oven compartment. The mass spectrometer was equipped with a Dual Agilent Jet Stream Electrospray Ionization (Dual AJS-ESI) source.

Chromatographic separation was achieved using a Waters XSelect™ HSS T3 (Milford, MA, USA) analytical column (2.1 × 150 mm, 2.5 μm) with a corresponding pre-column. The column oven temperature was set at 30 °C. A two-solvent mobile phase system was used: solvent A (5 mM ammonium acetate in 0.1% acetic acid aqueous solution) and solvent B (5 mM ammonium acetate and 0.1% acetic acid in a mix of methanol and acetonitrile, 80:20). The flow rate was maintained at 0.17 mL/min with a specific gradient elution program as follows: 0.00–1.00 min (40% B), 12.0 min (75% B), 20.0 min (85% B), 28.0–38.0 min (100% B), 40.0 min (75% B), returning to 40.0–45.0 min (40% B). Each chromatographic run included a 10 min column pre-equilibration at initial conditions (40% B). The multisampler was maintained at a temperature of 5 °C, and 5.00 μL of sample was injected for each analysis. Each biological replicate of all experimental conditions was injected twice as technical replicates.

The Agilent 6545 QTOF Mass Spectrometer operated in both positive and negative electrospray ionization (ESI) modes. Data acquisition was conducted using Agilent MassHunter Data Acquisition software (Version B:10.1.48), set to Auto MS/MS acquisition mode. The Dual AJS-ESI source conditions were meticulously configured: drying gas (N_2_) was maintained at 200 °C with a flow rate of 10.0 L/min; nebulizer gas (N_2_) pressure was set at 35 psi; and sheath gas (N2) was maintained at a temperature of 300 °C with a flow rate of 11.0 L/min. The instrument parameters were carefully adjusted, including capillary voltage at 2500 V; nozzle voltage at 100 V; fragmentor voltage at 100 V; skimmer voltage at 60 V; octupole RF voltage at 750 V; and collision energies were set at 0, 10, 20, and 40 eV. The mass spectrometer was tuned for a scan range of 50 to 1700 *m*/*z* at a rate of 5 spectra/s for MS scans, and 3 spectra/s for MS/MS scans with medium (~4 *m*/*z*) isolation width. The mass spectrometer was calibrated using the Agilent tuning mix solution before analysis to ensure mass accuracy throughout the data acquisition. Additionally, real-time mass correction and validation were carried out using the reference mass solution at *m*/*z* 922.0098 and *m*/*z* 1221.9906 for positive ionization mode and *m*/*z* 112.9855 and *m*/*z* 1033.9881 for negative ionization mode.

### 2.5. Data Processing, Statistical Analysis, and Metabolite Identification

Agilent MassHunter Data Acquisition software (Version: B.10.1.48) collected data from all cell studies (control, LPS-stimulated, and positive controls with triprolidine and zileuton) in both positive and negative ionization modes. The acquired data were stored in a (.d) file format. These files were subsequently analyzed using Agilent MassHunter Qualitative Analysis software (Version: B.10.0.1). This step involved assessing chromatographic peak shapes, retention times, and background noise in each mass spectrum.

Agilent MassHunter Profinder software (Version: B.10.0.2) was used for batch recursive feature extraction (i.e., molecular feature extraction and find by ion). The (.d) files obtained from the Agilent MassHunter Qualitative Analysis software were imported and processed based on their ionization mode (either positive or negative) across the four study conditions. The retention times across all runs were aligned using the internal standard employed for this study, and a minimum mass spectral peak height was set at 1200 counts. For molecular feature extraction, the extraction parameters included a minimum mass spectral peak height of 1500 counts, the allowed ion species of [M + H]^+^, [M + Na]^+^, and [M + NH_4_]^+^ for the positive ion mode, and [M − H]^−^ for the negative ion mode, the isotope model of common organic molecules without halogens, and the limit assigned charge states to a range of 1–2; the compound filters were set by default; the compound binning and alignment parameters included a retention time tolerance of 0.10% ± 0.30 min, and a mass tolerance of 20.00 ppm + 2.00 mDa; and the post-processing filters were set at an absolute height of at least 5000 counts for mass spectral peaks, a molecular feature extraction score of at least 75, and a minimum match of molecular feature at 75% (this meant a molecular feature must be present in 3 out of 4 replicate runs in each experimental condition to be included). For the find by ion, the matching tolerance and scoring parameters included a mass score of 100, isotope abundance and spacing scores of 60 and 50, respectively, and a retention score of 0; the EIC peak integration and filtering parameters included an absolute height of at least 7000 counts for chromatographic peaks; the spectrum extraction and centroiding parameters were set by default; and the post-processing filters included an absolute height of at least 7000 counts for chromatographic peak heights and a target score of at least 75.00. Finally, the data of all experimental groups by each ionization mode obtained from the operations of “molecular feature extraction” and “find by ion” were exported as profinder archive (.pfa) files from the Agilent MassHunter Profinder software. These (.pfa) files were uploaded to Agilent Mass Profiler Professional (MPP) software (Version: B.15.1.2) for statistical analysis and compound identification.

In MPP, the UHPLC-QTOF-MS data were normalized using the exogenous internal standard (PGE2-d4) in each sample to correct for signal fluctuations during instrumental analysis (since samples were previously normalized by cell count and total protein content, this was the third stage of sample normalization to ensure quantitative comparison of the data). Molecular features from each experimental condition were subjected to metabolite identification in MPP using the “ID Browser” tool, referencing the METLIN accurate-mass metabolites and lipids databases. Metabolites with identification scores below 75% were excluded to minimize false positives.

For each experimental condition, datasets containing two biological replicates and two technical replicates (i.e., BR1.TR1, BR1.TR2, BR2.TR1, and BR2.TR2) were exported from MPP as .csv files to create four independent datasets by averaging: Dataset 1 = (BR1.TR1 + BR2.TR1)/2, Dataset 2 = (BR1.TR2 + BR2.TR2)/2, Dataset 3 = BR1.TR1, and Dataset 4 = BR2.TR1. These independent datasets were then re-imported into MPP, and the median values for each condition were used to assess metabolite regulation. A one-way ANOVA followed by Tukey’s HSD test was applied, selecting for *p*-value, log2 fold change, and FDR of 0.05, using the Benjamini–Hochberg correction. Significantly regulated metabolites (*p* < 0.05, log2 fold change > 2, or fold change > 4) were identified between the following conditions: control vs. LPS-stimulated, LPS-stimulated vs. triprolidine-treated post-LPS, and LPS-stimulated vs. zileuton-treated post-LPS.

### 2.6. Principal Component Analysis and Pathway Enrichment Analysis

Principal component analysis (PCA) and pathway enrichment analysis were performed on the MetaboAnalyst 6.0 (available at https://www.metaboanalyst.ca/MetaboAnalyst/, accessed on 28 May 2024). In detail, the (.csv) files of sample replicates (i.e., two biological and two technical replicates) of each group from the same polarity (positive) were exported from Agilent MassHunter Profinder containing data like mass, retention time, and peak area. Replicate measurements’ (.csv) files were grouped together in a single folder for each condition (for, e.g., replicate files of control data into the control group folder). All the experimental condition groups, control, LPS-stimulated, and two distinct positive controls (triprolidine and zileuton), were merged into one (.zip) folder and uploaded to MetaboAnalyst. The mass tolerance of 0.025 Da and retention time tolerance of 30.0 s were set for processing the MS peak list data. The data were normalized using the IS reference feature (i.e., mass, retention time, and peak area), and data filtering was performed based on an “interquartile range” (IQR) of 5% to remove variables and increase the accuracy. The data were log-transformed (base 10) and auto-scaled. The processed data were then used to plot the 2D PCA plot. Similar steps were performed for negative polarity to plot another 2D PCA plot.

The (.csv) file containing the metabolite’s name and peak area for replicates of each experimental condition was exported from MPP for pathway analysis. Individual (.csv) files were generated to identify pathways regulated between two experimental groups (such as control vs. LPS-stimulated, LPS-stimulated vs. triprolidine positive control, and LPS-stimulated vs. zileuton positive control). This individual file was uploaded as a concentration table in the pathway analysis module of MetaboAnalyst, and the “ID type” and “data format” were set as “compound name” and “samples in column”, respectively. The data were log-transformed (base 10) and auto-scaled. The output parameters for pathway analysis were set to scatter plot (for testing significant features) for pathway analysis visualization, hypergeometric test for enrichment of the pathways identified, and relative-betweenness centrality for the topological analysis, and all the compounds in the pathway library were selected for metabolite ID reference. The *Mus musculus* (house mouse) organism was selected to obtain pathways from the KEGG database.

### 2.7. Semiquantitative Analysis

Relative semiquantitative analysis was performed using the (.d) files of the same polarity of the replicate samples of each experimental group, obtained from Agilent MassHunter Data Acquisition software and their corresponding (.cef) files with metabolite identities from the Agilent MPP software. The (.d) files were imported as samples, and the (.cef) file was imported as a method file for processing the semiquantitative analysis into Agilent MassHunter Quantitative analysis (Q-TOF) software (Version 10.2). The internal standard (PGE2-d4 in this case) was flagged as ISTD and then annotated as ISTD for all the metabolites with a concentration of 2.00 µg/mL in the compound setup section, and the metabolites were set as targets. The relative ISTD option was selected to carry out the relative quantitation of metabolites to the known internal standard. After validating the method, a semiquantitative analysis was executed based on an individual metabolite’s peak area to the IS’s peak area multiplied by the IS concentration. The results were exported to a Microsoft Excel sheet for reporting.

## 3. Results

### 3.1. Validation of the UHPLC-QTOF-MS-Based Method

In this work, a UHPLC-QTOF-MS-based method was developed and validated for untargeted and targeted metabolomic analysis of murine mast cell (MC/9) samples under various experimental conditions, including a negative control (untreated), an LPS-stimulated sample, and two drug-treated samples (i.e., triprolidine and zileuton) post-LPS-stimulation as the positive controls. This method employed an exogenous stable heavy isotope (PGE2-d4) as the internal standard to assess the matrix effects, mass and retention alignment, relative quantitation, and cross-comparison of the data between the experimental conditions. Figure 1 shows the representative total ion current chromatograms (TICs) of untargeted metabolomic profiling of cell samples under various experimental conditions, whereas Figure 2 shows the representative extracted ion chromatograms (EICs) of targeted metabolomic analysis of some individual metabolites in cell extracts.

The effects of the sample matrix are crucial when using UHPLC-QTOF-MS-based methods in metabolomics studies [18]. Therefore, the matrix effects in both positive and negative ionization modes were determined by spiking an internal standard PGE2-d4 into various cell samples. The matrix effects expressed as matrix factors were determined by the mean peak area of PGE2-d4 at a specified concentration in an extracted cell sample from an experimental condition over the mean peak area of PGE2-d4 at the same concentration in the mobile phase. Table 1 indicated the matrix factors were 0.98, 0.82, 0.92, and 0.97 for negative electrospray ionization mode and 1.12, 1.05, 1.00, and 1.00 for positive electrospray ionization mode for the untreated control, LPS-stimulated, triprolidine-treated, and zileuton-treated cell samples, respectively. The matrix effects were corrected in this study since the peak area ratios of metabolites vs. IS were used in data analysis.

The reproducibility of the untargeted metabolomics profiling was assessed by multivariate analysis and visualized by the principal component analysis (PCA) score plots. As shown in Figure 3, the close grouping of replicate measurements (i.e., two biological and two technical replicates) for samples of each experimental group in the PCA score plot indicated excellent precision of the UHPLC-QTOF-MS-based method. These grouping clusters represented distinct metabolomic profiles of the experimental groups. The PCA plot showed that the principal component 1 (PC1) and principal component 2 (PC2) scores were 51.2% and 32.2%, respectively, accounting for 83.4% of the total variance for data acquired in positive ionization mode, whereas for negative ionization mode, the PC1 and PC2 scores were 64.5% and 24.1%, respectively, accounting for 88.6% of the total variance. As shown in Figure 3, a vector shift from the control to the LPS-stimulated group indicated a significant metabolomic alteration under allergic stimulation. In contrast, vectors for triprolidine- and zileuton-treated groups showed a metabolomic profile shift from the LPS-stimulated group back toward the control group, suggesting a reversal from the LPS-stimulated profile. Furthermore, the reproducibility of the targeted metabolomic analysis was demonstrated by the coefficients of variation (CVs) of the metabolites in global semiquantitative analysis. Table 3 shows the CVs of the replicate measurements of 44 regulated metabolites in the four experimental groups. Compared to the recommended values (at least 70% metabolites at CVs ≤ 15%) [19], there were 88.6–97.7% of metabolites measured with CVs ≤ 15% from the four experimental groups, indicating good reproducibility of the method for targeted metabolomic analysis.

### 3.2. Metabolite Identification and Semiquantitative Analysis

The untargeted metabolomic profiling of samples from four experimental conditions (control (untreated), LPS-stimulated, triprolidine-treated post-LPS, and zileuton-treated post-LPS) yielded 3435 molecular features. These features were subjected to metabolite identification using the “ID Browser” tool within MPP and the METLIN accurate-mass metabolite and lipid databases.

To ensure confidence in the identification of metabolites, molecular features extracted from a sample were analyzed using both MS and MS/MS data, which were matched against the METLIN database. METLIN libraries include MS and MS/MS spectra for standards, enabling a robust comparison. Metabolites were annotated with a passing score threshold of 75.0, ensuring a high confidence level in metabolite identifications.

Prior to statistical analysis, four independent datasets were generated by averaging technical replicates from each experimental condition, which included two biological replicates and two technical replicates per condition. A one-way ANOVA followed by Tukey’s HSD test (with *p*-value < 0.05, log2 fold change > 2, and false discovery rate (FDR) of 0.05) identified 44 significantly regulated metabolites across the following comparisons: control vs. LPS-stimulated, LPS-stimulated vs. triprolidine-treated post-LPS, and LPS-stimulated vs. zileuton-treated post-LPS (Table 2). These forty-four metabolites included twenty-two amino acids and other organic acids, two peptides, six leukotrienes, six lipids and ethers, five thromboxanes, and three other metabolites: histamine, aminofructose-6-phosphate, and lipoxin C4. The significantly regulated metabolites were further subjected to semiquantitative analysis, with their concentrations determined relative to the known concentration of the IS. These concentrations, ranging from picomolar to tens of nanomolar, are summarized in Table 3.

**Table 2 biomolecules-15-00109-t002:** The 44 significantly regulated metabolites identified by UHPLC-QTOF-MS-based untargeted metabolomics.

Metabolites	Chemical Formula	*m*/*z*	Ion Species	METLIN ID	PubChem ID
** *Amino Acids* **					
L-Arginine	C_6_H_14_N_4_O_2_	175.119	[M + H]^+^	13	6322
L-Asparagine	C_4_H_8_N_2_O_3_	133.0608	[M + H]^+^	14	6267
L-Aspartic Acid	C_4_H_7_NO_4_	134.0448	[M + H]^+^	15	5960
L-Glutamine	C_5_H_10_N_2_O_3_	164.103	[M + NH_4_]^+^	18	5961
L-Histidine	C_6_H_9_N_3_O_2_	154.0622	[M − H]^−^	21	6274
L-Isoleucine	C_6_H_13_NO_2_	132.1019	[M + H]^+^	23	6306
L-Phenylalanine	C_9_H_11_NO_2_	166.0863	[M + H]^+^	28	6140
L-Proline	C_5_H_9_NO_2_	116.0706	[M + H]^+^	29	145742
L-Serine	C_3_H_7_NO_3_	104.0353	[M − H]^−^	30	5951
L-Tryptophan	C_11_H_12_N_2_O_2_	205.0972	[M + H]^+^	33	6305
L-Valine	C_5_H_11_NO_2_	118.0863	[M + H]^+^	35	6287
** *Organic Acids* **					
(S)-beta-Methylindolepyruvate *	C_12_H_11_NO_3_	240.0631	[M+Na]^+^	66077	440163
2-Hydroxy-4-hydroxymethylbenzalpyruvate	C_11_H_10_O_5_	223.0601	[M + H]^+^	69802	5282333
3-Methylbutyl 2-oxopropanoate	C_8_H_14_O_3_	159.1016	[M + H]^+^	93029	537682
8S,15S-diHPETE *	C_20_H_32_O_6_	386.2537	[M + NH_4_]^+^	75024	52921888
9-HpETE	C_20_H_32_O_4_	359.2193	[M + Na]^+^	36294	5283173
Enol-phenylpyruvate	C_9_H_8_O_3_	165.0546	[M + H]^+^	6098	641637
Dinoprost (PGF2α)	C_20_H_34_O_5_	355.2479	[M + H]^+^	36085	483926742
Epoprostenol (PGI2)	C_20_H_32_O_5_	353.2323	[M + H]^+^	36155	5282411
Retinoic Acid	C_20_H_28_O_2_	299.2017	[M − H]^−^	2277	444795
Arachidonic Acid	C_20_H_32_O_2_	303.233	[M − H]^−^	193	444899
S-(4-Bromophenyl)-mercaptopyruvate *	C_9_H_6_BrO_3_S	271.9148	[M − H]^−^	66147	5459980
** *Ethers* **					
Dihydroxyacetone Phosphate Acyl Ester	C_4_H_7_O_7_P	199.0002	[M + H]^+^	62468	6857386
Ethyl Pyruvate	C_5_H_8_O_3_	117.0546	[M + H]^+^	87833	12041
** *Leukotrienes* **					
Leukotriene A4 (LTA4)	C_20_H_30_O_3_	319.2268	[M + H]^+^	3449	5280383
Leukotriene D4 (LTD4)	C_25_H_40_N_2_O_6_S	519.2499	[M + Na]^+^	3583	5280878
Leukotriene E4 (LTE4)	C_23_H_37_NO_5_S	457.2731	[M + NH_4_]^+^	3536	5280879
Leukotriene B3 (LTB3) *	C_20_H_34_O_4_	339.253	[M + H]^+^	43393	6439476
Leukotriene B4 (LTB4)	C_20_H_32_O_4_	337.2373	[M − H]^+^	406	5280492
Leukotriene B5 (LTB5)	C_20_H_30_O_4_	352.2482	[M + NH_4_]^+^	36241	5283125
** *Lipids* **					
Cer(d18:1/14:0) *	C_32_H_63_NO_3_	510.4881	[M + H]^+^	41564	5282310
Cer(d18:1/16:0)	C_34_H_67_NO_3_	538.5194	[M + H]^+^	83706	5283564
CerP(d18:1/20:0) *	C_38_H_76_NO_6_P	691.5749	[M + NH_4_]^+^	41578	5283584
DHAP(18:0)	C_21_H_41_O_7_P	454.2928	[M + NH_4_]^+^	61995	440127
** *Peptides* **					
Glutathione (GSH) *	C_10_H_17_N_3_O_6_S	308.0911	[M + H]^+^	44	124886
Oxiglutatione (GSSG) *	C_20_H_32_N_6_O_12_S_2_	611.1446	[M − H]^−^	45	65359
** *Thromboxanes* **					
Thromboxane	C_20_H_40_O	319.2971	[M + Na]^+^	415	114873
Thromboxane A2 (TXA2)	C_20_H_32_O_5_	353.2323	[M + H]^+^	63351	5280497
Thromboxane A3 (TXA3)	C_20_H_30_O_5_	351.2166	[M + H]^+^	36260	5283139
Thromboxane B1 (TXB1) *	C_20_H_36_O_6_	373.2585	[M + H]^+^	36262	16061114
Thromboxane B2 (TXB2)	C_20_H_34_O_6_	371.2428	[M + H]^+^	422	5283137
** *Others* **					
Histamine	C_5_H_9_N_3_	112.0869	[M + H]^+^	68	774
Aminofructose 6-phosphate	C_6_H_14_NO_8_P	277.0795	[M + NH_4_]^+^	63735	443711
Lipoxin C4 *	C_30_H_47_N_3_O_10_S	642.3055	[M + H]^+^	45970	9548805

* This metabolite was not recognized by MetaboAnalyst for pathway analysis.

**Table 3 biomolecules-15-00109-t003:** The concentrations of 44 significantly regulated metabolites in mast cells under various experimental conditions by UHPLC-QTOF-MS-based targeted metabolomics.

Metabolite	Mean Concentration (nM) ± Standard Deviation (% CV) *(n* = 4)			
	Control	LPS-Stimulated	Triprolidine-Treated Post-LPS	Zileuton-Treated Post-LPS
(S)-beta-Methylindolepyruvate	2.27 × 10 ± 0.01 (3)	2.92 × 10 ± 0.00 (1)	6.72 × 10 ± 0.01 (1)	9.42 × 10 ± 0.00 (0)
2-Hydroxy-4-hydroxymethylbenzalpyruvate	1.98 × 10 ± 0.02 (9)	1.62 × 10 ± 0.00 (0)	4.31 × 10^−2^ ± 0.00 (11)	3.82 × 10^−2^ ± 0.00 (7)
3-Methylbutyl 2-oxopropanoate	5.22 × 10 ± 0.01 (2)	4.44 × 10 ± 0.01 (2)	2.79 × 10 ± 0.02 (3)	8.94 × 10^−2^ ± 0.01 (6)
8S,15S-diHPETE	7.86 × 10 ± 0.01 (18)	1.9 ± 0.2 (10)	1.15 ± 0.17 (15)	4.74 × 10 ± 0.03 (6)
9-HpETE	5.42 × 10^−2^ ± 0.00 (5)	2.11 ± 0.01 (0)	8.9 × 10 ± 0.1 (5)	1.56 × 10 ± 0.01 (7)
Aminofructose 6-phosphate	1.53 × 10 ± 0.14 (9)	1.76 × 10 ± 0.02 (11)	4.78 × 10 ± 0.00 (0)	3.78 × 10 ± 0.01 (3)
Arachidonic Acid	2.1 × 10 ± 1.3 (6)	9.6 × 10 ± 8.2 (9)	3.33 × 10 ± 1.55 (5)	3.06 × 10 ± 0.06 (0)
Cer(d18:1/14:0)	3.96 × 10^−2^ ± 0.00 (2)	1.51 × 10 ± 0.00 (1)	5.35 × 10^−2^ ± 0.00 (3)	4.00 × 10^−2^ ± 0.00 (5)
Cer(d18:1/16:0)	5.92 × 10 ± 0.01 (1)	3.56 ± 0.01 (0)	6.69 × 10 ± 0.00 (1)	7.45 × 10 ± 0.02 (2)
CerP(d18:1/20:0)	4.0 × 10 ± 0.5 (12)	1.1 × 10 ± 0.1 (1)	9.8 ± 0.1 (1)	1.1 × 10 ± 0.5 (5)
DHAP(18:0)	4.45 × 10 ± 0.01 (1)	3.61 ± 0.02 (0)	2.43 ± 0.02 (1)	3.43 ± 0.01 (0)
Dihydroxyacetone Phosphate Acyl Ester	3.08 × 10 ± 0.00 (2)	4.23 × 10 ± 0.00 (1)	9.40 × 10 ± 0.03 (4)	2.07 ± 0.01 (1)
Enol-phenylpyruvate	4.36 × 10^−3^ ± 0.00 (9)	5.37 × 10^−3^ ± 0.00 (0)	1.07 × 10^−2^ ± 0.00 (0)	5.37 × 10^−2^ ± 0.00 (0)
Ethyl Pyruvate	1.47 × 10^−3^ ± 0.00 (11)	7.78 × 10^−3^ ± 0.00 (11)	2.68 × 10^−3^ ± 0.00 (8)	8.76 × 10^−3^ ± 0.00(10)
Glutathione (GSH)	2.09 × 10^−2^ ± 0.00 (2)	5.10 × 10 ± 0.01 (1)	4.18 × 10^−2^ ± 0.00 (2)	5.09 × 10 ± 0.02 (3)
Oxiglutatione (GSSG)	4.76 × 10 ± 0.00 (8)	2.0 ± 0.2 (11)	9.11 × 10 ± 0.01 (1)	1.28 ± 0.01 (1)
Histamine	2.4 × 10 ± 0.2 (8)	1.2 × 10 ± 0.1 (1)	3.1 ± 0.3 (8)	1.09 × 10 ± 0.15 (1)
L-Arginine	1.58 × 10^−2^ ± 0.00 (10)	9.7 × 10 ± 0.1 (12)	1.92 × 10 ± 0.01 (3)	6.42 × 10^−2^ ± 0.00 (7)
L-Asparagine	1.06 × 10^−2^ ± 0.00 (3)	4.4 × 10 ± 0.1 (12)	2.19 × 10 ± 0.02 (8)	3.64 × 10 ± 0.01 (2)
L-Aspartic Acid	5.99 × 10^−2^ ± 0.00 (0)	7.98 × 10^−2^ ± 0.00 (1)	9.01 × 10^−2^ ± 0.00 (2)	1.03 × 10 ± 0.00 (2)
Leukotriene A4 (LTA4)	5.70 × 10^−2^ ± 0.00 (5)	5.11 × 10 ± 0.00 (1)	3.61 × 10 ± 0.01 (2)	2.21 × 10 ± 0.01 (3)
Leukotriene B3 (LTB3)	2.4 × 10 ± 0.2 (10)	1.3 × 10 ± 0.4 (3)	6.4 ± 0.1 (1)	2.7 ± 0.1 (2)
Leukotriene B4 (LTB4)	2.67 × 10^−3^ ± 0.00 (1)	5.5 ± 0.1 (1)	3.9 ± 0.1 (1)	6.03 × 10 ± 0.01 (2)
Leukotriene B5 (LTB5)	2.60 × 10 ± 0.01 (5)	1.4 ± 0.1 (8)	7.16 × 10 ± 0.01 (2)	2.71 × 10 ± 0.03 (12)
Leukotriene D4 (LTD4)	2.08 × 10^−2^ ± 0.00 (7)	7.07 × 10 ± 0.04 (6)	5.30 × 10 ± 0.00 (0)	4.77 × 10^−2^ ± 0.01 (16)
Leukotriene E4 (LTE4)	2.35 × 10^−2^ ± 0.00 (1)	4.42 × 10 ± 0.01 (2)	3.82 × 10 ± 0.00 (0)	4.00 × 10^−2^ ± 0.01 (18)
L-Glutamine	3.69 × 10 ± 0.01 (2)	1.6 ± 0.1 (4)	4.1 × 10 ± 0.1 (15)	4.9 × 10 ± 0.1 (17)
L-Histidine	7.13 × 10 ± 0.03 (4)	1.76 × 10 ± 0.02 (9)	3.1 × 10 ± 0.0 (5)	2.79 × 10 ± 0.01 (1)
Lipoxin C4	2.9 × 10 ± 0.1 (23)	1.72 × 10 ± 0.01 (8)	1.04 × 10 ± 0.01 (9)	7.71 × 10^−2^ ± 0.01 (11)
L-Isoleucine	3.71 × 10 ± 0.03 (1)	5.5 ± 0.0 (0)	6.9 ± 0.1 (2)	7.88 ± 0.02 (0)
L-Phenylalanine	7.21 × 10 ± 0.01 (2)	8.7 ± 0.4 (5)	3.1 ± 0.2 (7)	3.29 ± 0.03 (1)
L-Proline	1.40 × 10 ± 0.02 (11)	5.97 × 10 ± 0.03 (5)	2.18 × 10 ± 0.04 (16)	2.36 × 10 ± 0.01 (5)
L-Serine	4.49 × 10^−4^ ± 0.00 (7)	3.59 × 10^−2^ ± 0.00 (1)	1.78 × 10^−2^ ± 0.00 (4)	2.40 × 10^−2^ ± 0.00 (2)
L-Tryptophan	6.67 × 10 ± 0.01 (1)	8.44 × 10 ± 0.04 (4)	1.6 ± 0.1 (3)	2.78 ± 0.04 (1)
L-Valine	2.26 × 10 ± 0.00 (1)	9.8 × 10 ± 0.1 (12)	3.65 × 10 ± 0.02 (4)	3.25 × 10 ± 0.02 (6)
Dinoprost (PGF2α)	1.09 × 10 ± 0.04 (4)	1.1 × 10 ± 1.7 (16)	4.1 ± 0.7 (17)	3.8 ± 0.3 (8)
Epoprostenol (PGI2)	1.92 × 10 ± 0.00 (1)	9.60 × 10 ± 0.02 (3)	4.65 × 10 ± 0.02 (4)	4.73 × 10 ± 0.00 (1)
Retinoic Acid	9.1 × 10 ± 0.6 (6)	3.90 × 10 ± 3 (8)	2.7 × 10 ± 0.5 (2)	2.7 × 10 ± 0.3 (1)
S-(4-Bromophenyl)-mercaptopyruvate	7.20 × 10^−3^ ± 0.00 (10)	1.02 × 10^−2^ ± 0.00 (11)	9.09 × 10^−3^ ± 0.00 (3)	3.37 × 10^−3^ ± 0.00 (3)
Thromboxane	6.26 × 10^−3^ ± 0.00 (9)	1.59 × 10^−2^ ± 0.00 (7)	1.82 × 10^−2^ ± 0.00 (13)	3.63 × 10^−2^ ± 0.00 (9)
Thromboxane A2 TXA2)	7.30 × 10^−4^ ± 0.00 (10)	7.01 × 10^−2^ ± 0.00 (6)	3.53 × 10^−3^ ± 0.00 (0)	5.84 × 10^−3^ ± 0.00 (21)
Thromboxane A3 (TXA3)	3.63 × 10^−3^ ± 0.00 (11)	3.98 × 10^−2^ ± 0.00 (8)	1.87 × 10^−2^ ± 0.00 (9)	1.89 × 10^−2^ ± 0.00 (22)
Thromboxane B1 (TXB1)	5.80 × 10^−3^ ± 0.00 (0)	3.67 × 10^−2^ ± 0.00 (9)	8.07 × 10^−3^ ± 0.00 (15)	1.06 × 10 ± 0.01 (10)
Thromboxane B2 (TXB2)	4.65 × 10^−4^ ± 0.00 (14)	9.76 × 10^−3^ ± 0.00 (9)	6.01 × 10^−4^ ± 0.00 (7)	8.26 × 10^−4^ ± 0.00 (13)

### 3.3. Metabolite Regulation and Pathway Analysis

Figure 4 visualizes the contents of 44 significantly regulated metabolites across the four experimental conditions, highlighting statistical comparisons between control vs. LPS-stimulated, LPS-stimulated vs. triprolidine-treated post-LPS, and LPS-stimulated vs. zileuton-treated post-LPS with the detailed *p*-values and log2 fold change values supplemented in Appendix A (Appendix A). This figure emphasizes the metabolic alterations in mast cells under conditions of induced allergic rhinitis and demonstrates the therapeutic effects of the positive control drugs, triprolidine and zileuton.

In the LPS-stimulated group, 39 of the 44 metabolites were upregulated compared to the control group. The exceptions were aminofructose 6-phosphate, l-histidine, 2-hydroxy-4-hydroxymethylbenzalpyruvate, lipoxin C4, and 3-methylbutyl 2-oxopropanoate, which were downregulated. Compared to the LPS-stimulated group, the triprolidine-treated group showed downregulation in 35 of the 44 metabolites. However, aminofructose 6-phosphate, l-aspartic acid, (S)-beta-methylindolepyruvate, dihydroxyacetone phosphate acyl ester, enol pyruvate, l-histidine, l-isoleucine, thromboxane, and l-tryptophan were upregulated. Similarly, 31 of the 44 metabolites were downregulated in the zileuton-treated group compared to the LPS-stimulated group. Two metabolites, CerP(d18:1/20:0) and glutathione, remained unchanged, while aminofructose 6-phosphate, l-aspartic acid, (S)-beta-methylindolepyruvate, dihydroxyacetone phosphate acyl ester, enol pyruvate, ethyl pyruvate, l-histidine, l-isoleucine, thromboxane, thromboxane B1, and l-tryptophan, were upregulated.

Pathway analysis was performed using MetaboAnalyst with the KEGG pathway database to elucidate the impact of metabolite regulation. Of the 44 significantly regulated metabolites, 10 were not identified by MetaboAnalyst. Therefore, the pathway analysis was based on 34 metabolites (see Table 2). Figure 5 highlights pathways with an impact score ≥ 0.2 and a log10(p) ≥ 2 (equivalent to a *p*-value ≤ 0.01), indicating significantly altered metabolomic profiles under the experimental conditions. Specifically, six pathways—phenylalanine, tyrosine, and tryptophan biosynthesis; histidine metabolism; arachidonic acid metabolism; phenylalanine metabolism; sphingolipid metabolism; and glycine, serine, and threonine metabolism—were significantly altered in mast cells stimulated by LPS (Figure 5a). This suggests a complex and multifaceted cellular response, likely indicative of an inflammatory reaction. In contrast, treating LPS-stimulated mast cells with triprolidine (Figure 5b) significantly affected three pathways: histidine metabolism, sphingolipid metabolism, and glycine, serine, and threonine metabolism. This indicates that triprolidine not only acts as an antihistamine but also has broader effects on cellular metabolism, contributing to regulating the inflammatory response in mast cells. Similarly, treating LPS-stimulated mast cells with zileuton (Figure 5c) significantly modulated two pathways: arachidonic acid and sphingolipid metabolism. This highlights zileuton’s role in not only inhibiting a specific inflammatory pathway (leukotriene synthesis) but also broadly influencing cell signaling and inflammatory responses through its impact on sphingolipid metabolism.

## 4. Discussion

UHPLC-QTOF-MS-based methods are superior for cell metabolomics studies, surpassing the capabilities of traditional targeted metabolite analysis and protein-centric methodologies. UHPLC provides high-resolution separation of metabolites, allowing for the analysis of complex biological samples with a wide range of polarities and molecular structures. QTOF-MS offers high mass accuracy and resolution, which is essential for identifying and quantifying a vast array of metabolites, including those at low concentrations. The combination of UHPLC and QTOF-MS enables the detection of low-abundance metabolites with high sensitivity, which is critical in cell metabolomics, where metabolites can vary significantly in concentration. The high specificity of QTOF-MS ensures accurate identification of metabolites, reducing the chances of false positives. Furthermore, UHPLC-QTOF-MS-based methods can perform both untargeted (discovery) and targeted (quantitative) metabolomics, providing a comprehensive view of the metabolome. Untargeted analysis allows the discovery of novel metabolites and pathways, while targeted analysis can validate and quantify known metabolites.

The advantages of UHPLC-QTOF-MS-based methods over traditional targeted metabolite analysis include broader metabolite coverage, high throughput, and efficiency. UHPLC-QTOF-MS-based methods allow for simultaneously detection and quantification of a wide range of metabolites, offering a more holistic view of the metabolome. They can process multiple metabolites rapidly with high reproducibility, making them suitable for large-scale studies and high-throughput screening. In contrast, traditional targeted metabolite analysis focuses on a predefined set of metabolites, often requiring separate analyses for different metabolite groups. This limitation reduces the scope of the study and decreases efficiency.

Compared to protein-centric methodologies such as Western blot and ELISA assays, UHPLC-QTOF-MS-based methods directly measure metabolites, providing a more immediate and accurate reflection of cellular metabolism and biochemical pathways. They also offer a broader dynamic range and higher quantitative accuracy, which is critical for detecting subtle changes in metabolite levels. Western blot and ELISA measure protein levels, which are indirect indicators of metabolic states and often suffer from a limited dynamic range and issues with antibody specificity and sensitivity. Metabolomics with UHPLC-QTOF-MS provides insights into metabolic pathways and their regulation, enabling a deeper understanding of cellular processes. In contrast, protein-centric methods focus on protein abundance and modifications, which may not directly correlate with metabolic changes.

The downsides of using UHPLC-QTOF-MS for metabolomic investigations include the following: (i) Cost and complexity: The instrumentation and maintenance are expensive, and the technology demands specialized expertise for operation and data analysis. (ii) Matrix effects: Complex biological samples can introduce matrix effects, potentially compromising metabolite quantification accuracy. These issues can be addressed through the use of internal standards and properly prepared samples. (iii) Data complexity: The large datasets generated require advanced bioinformatics tools for processing and interpretation, which can be time-intensive. Despite these limitations, UHPLC-QTOF-MS remains a powerful tool for studying cell metabolism and drug efficacy. It offers high sensitivity, broad metabolite coverage, and the ability to elucidate complex biochemical pathways. While the method has challenges, its advantages make it highly valuable for advancing research in allergic rhinitis and evaluating therapeutic interventions.

While multiple cell types, including eosinophils, basophils, epithelial cells, and lymphocytes, contribute to allergic rhinitis, mast cells are central to its pathophysiology. They initiate and propagate allergic inflammation by releasing key mediators such as histamine and lipid molecules, which drive the characteristic symptoms of allergic rhinitis. This makes mast cells the most suitable model for studying the disease. Although human mast cell lines like HMC-1 and LAD2 closely mimic human disease, their slow doubling time and demanding growth conditions make them less practical for initial method development and validation. Instead, mouse mast cells (MC/9) were chosen due to their faster growth, easier cultivation, and ability to produce large cell quantities.

MC/9 cells are a well-established murine mast cell line widely used in immunological research. They share many features with primary mast cells and exhibit pathophysiology sufficiently representative of human mast cell behavior, making them a reliable model for studying allergic rhinitis [20]. Mast cells, including MC/9 cells, can be activated by various pathogens and allergens through receptors such as the high-affinity IgE receptor (FcεRI), Toll-like receptors (TLRs), and others [21]. Mast cells play critical roles in both innate and adaptive immunity. They act as first responders to pathogens (innate immunity) and modulate adaptive immune responses through interactions with other immune cells [22]. Upon activation, mast cells degranulate, releasing a variety of pre-stored and newly synthesized inflammatory mediators, including histamine, cytokines, and leukotrienes [23]. These inflammatory mediators are central to the pathophysiology of allergic rhinitis, contributing to symptoms such as nasal congestion, itching, sneezing, and rhinorrhea [24].

This study investigates cell metabolomic alterations associated with allergic responses. We identify metabolic changes in response to pathogen (LPS) exposure and treatment with established anti-allergic drugs (triprolidine and zileuton). Additionally, we examine how these metabolic changes influence signal transduction pathways critical for mast cell activation and degranulation, confirming the therapeutic targets of the established drugs.

LPS, a component of the outer membrane of Gram-negative bacteria, is recognized by Toll-like receptor 4 (TLR4) on the surface of MC/9 cells, facilitated by the co-receptor MD-2 and the accessory protein CD14 [25,26]. The binding of LPS to TLR4 initiates a complex signaling cascade involving the recruitment of adaptor proteins MyD88 (myeloid differentiation primary response 88) and TRIF (TIR-domain-containing adapter-inducing interferon-β) [25]. MyD88-dependent pathways predominantly lead to the activation of NF-κB (nuclear factor kappa B) and MAPK (mitogen-activated protein kinase) pathways, while TRIF-dependent pathways are associated with IRF3 activation and type I interferon production [25]. Activation of these pathways results in the transcription and secretion of various pro-inflammatory cytokines and chemokines.

MC/9 cells produce and release tumor necrosis factor-alpha (TNF-α), interleukin-6 (IL-6), and interleukin-1 beta (IL-1β) in response to LPS stimulation, which is crucial in mediating inflammatory responses and recruiting other immune cells [27,28]. Although LPS is less potent in inducing degranulation than allergens that crosslink IgE receptors, it can still trigger the release of pre-stored mediators in mast cells, including histamine, proteases, and other bioactive compounds contributing to the inflammatory response [27]. LPS activation also leads to the upregulation of co-stimulatory molecules such as CD40, CD80, and CD86 on the surface of MC/9 cells, enhancing their ability to interact with T cells and other immune cells, thereby facilitating the adaptive immune response [27]. The overall functional state of MC/9 cells is modulated upon LPS activation, including enhanced antigen presentation capabilities and altered cytokine profiles, influencing interactions with dendritic cells, B cells, and other immune system components [21].

In this study, we detected and identified significantly regulated metabolites in the MyD88-dependent pathways (Figure 6). In the NF-κB pathway, transcription factors AP1 (activator protein 1) and SP1 (specificity protein 1) enhance the transcriptional activity of the HDC (histidine decarboxylase) gene by binding to its promoter region, leading to increased conversion of histidine into histamine, an inflammatory mediator in allergic reactions [29]. Our experimental data (Figure 4, Table 3) support this, showing strongly upregulated histamine and downregulated L-histidine when mast cells were stimulated by LPS. In the MAPK pathway, cPLA2 (cyclic phospholipase A2) is phosphorylated and activated [30], leading to the production of arachidonic acid [31]. Arachidonic acid is a substrate for COX-2 (cyclooxygenase-2) and ALOX5 (arachidonate 5-lipoxygenase) to synthesize various inflammatory mediators. Our experimental data (Figure 4, Table 3) also show that the upregulation of arachidonic acid leads to a cascade upregulation of HPETEs (hydroperoxyeicosatetraenoic acids: 8S,15S-diHPETE, 9-HpETE), leukotrienes (A4, B3, B4, B5, D4, E4), prostaglandins (PGF2α, PGI2), and thromboxanes (thromboxane, A2, A3, B1, B2). These metabolites play significant roles in inflammatory responses and are observed in the LPS-stimulated mast cells.

When LPS-stimulated mast cells were treated with zileuton, an inhibitor of leukotriene synthesis that works by deactivating the ALOX5 enzyme, leukotrienes were downregulated, as expected, and histamine levels were almost unaffected (Figure 4, Table 3). Leukotrienes are inflammatory mediators that contribute to allergic responses, including allergic rhinitis, causing bronchoconstriction, mucus production, and increased vascular permeability, leading to symptoms like nasal congestion and sinus pressure [32]. Treating with zileuton, which inhibits leukotriene production, shows the role of leukotrienes in allergic rhinitis and the potential effectiveness of leukotriene inhibitors as therapeutic agents. When LPS-stimulated mast cells were treated with triprolidine, an antihistamine that blocks the H1 receptor, histamine levels were notably downregulated, and L-histidine, the histamine precursor, exhibited an inverse trend to histamine, suggesting shifts in the histamine synthesis pathway. This study utilized two well-known pharmacological agents, triprolidine and zileuton, as positive controls, allowing for a comprehensive investigation of the cellular and molecular mechanisms underlying allergic rhinitis. It also provided a means to evaluate the therapeutic effectiveness of targeting different mediators (histamine and leukotrienes) in modulating allergic responses.

Pathway analysis revealed significant alterations in six metabolic pathways in mast cells stimulated by LPS (Figure 5a), suggesting a coordinated metabolic reprogramming in response to LPS stimulation. This reprogramming likely supports the production of inflammatory mediators, enhances inflammation-related signaling processes, and adjusts cellular metabolism to meet the energetic and biosynthetic demands of the immune response. Here is an overview of the potential changes in each pathway: (1) Phenylalanine, tyrosine, and tryptophan biosynthesis: These amino acids are precursors to neurotransmitters such as dopamine, norepinephrine, and serotonin. Alterations in this pathway may indicate increased synthesis of these compounds, which can modulate immune responses and inflammation. (2) Histidine metabolism: Histidine is a precursor to histamine, a well-known mediator of allergic responses and inflammation. Changes in this pathway suggest increased histamine production, contributing to the inflammatory response. (3) Arachidonic acid metabolism: Arachidonic acid is crucial for the production of eicosanoids (prostaglandins, thromboxanes, and leukotrienes), which are potent mediators of inflammation. Alterations in this pathway indicate enhanced production of these inflammatory mediators. (4) Phenylalanine metabolism: This pathway is connected to the production of tyrosine and downstream neurotransmitters and bioactive molecules. Changes here further support the involvement of bioactive amines in the immune response. (5) Sphingolipid metabolism: Sphingolipids are essential components of cell membranes and play a role in signaling processes, including stress responses and inflammation. Alterations in this pathway suggest changes in cell membrane dynamics and signaling that are important for inflammatory responses. (6) Glycine, serine, and threonine metabolism: These amino acids are involved in various metabolic processes, including synthesizing proteins and nucleotides. Changes in this pathway reflect shifts in energy metabolism and biosynthetic activities in response to inflammatory stimuli.

Treating LPS-stimulated mast cells with triprolidine (Figure 5b) modulates the immune response by affecting histamine signaling. This modulation leads to significant changes in histidine metabolism, reducing histamine levels or activity. These primary effects cascade into alterations in sphingolipid metabolism, impacting membrane dynamics and cell signaling, and glycine, serine, and threonine metabolism, affecting cell energy balance and biosynthesis. These changes suggest a shift towards a less inflammatory and more regulated metabolic state in response to triprolidine. Here is an analysis of the observed alterations: (1) Histidine metabolism: Triprolidine, an antihistamine, inhibits histamine receptor activity. This inhibition might trigger feedback mechanisms that alter histidine metabolism, potentially reducing the conversion of histidine to histamine or affecting overall histamine levels in the cells. The reduced histamine activity could impact various downstream processes, including inflammatory responses and cell signaling, leading to a rebalancing of metabolic flux through the histidine pathway. (2) Sphingolipid metabolism: Triprolidine’s effect on histamine receptors, which are G-protein-coupled receptors, might indirectly influence sphingolipid metabolism. Sphingolipids, such as Cer(d18:1/16:0) and Cer(d18:1/14:0), are vital for membrane structure and signaling. Changes in this pathway might reflect alterations in cell membrane dynamics and signal transduction, possibly due to reduced histamine-mediated signaling. Sphingolipids are also involved in the cellular stress response. By modulating histamine activity, triprolidine could impact sphingolipid-mediated stress response pathways, affecting cell survival and inflammatory processes. (3) Glycine, serine, and threonine metabolism: These amino acids play roles in numerous biosynthetic and metabolic processes. Triprolidine’s effect on histamine signaling might lead to changes in the cellular demand for these amino acids, altering their metabolism. Alterations in this pathway could indicate changes in energy metabolism and the synthesis of nucleotides and other molecules necessary for cell growth and repair. This could be a compensatory mechanism in response to the altered inflammatory environment due to triprolidine treatment. Therefore, triprolidine not only acts as an antihistamine but also has broader effects on cellular metabolism, contributing to regulating the inflammatory response in mast cells.

Zileuton treatment of LPS-stimulated mast cells (Figure 5c) significantly modulates arachidonic acid and sphingolipid metabolisms, indicating its specific effects on the inflammatory response and cellular signaling. Here is a detailed analysis of the observed effects: (1) Arachidonic acid metabolism: LPS stimulation likely upregulates arachidonic acid metabolism, leading to increased production of eicosanoids, such as leukotrienes, prostaglandins, and thromboxanes (Figure 4, Table 3), which are potent inflammatory mediators that amplify the immune response. Zileuton, an ALOX5 inhibitor, specifically inhibits the synthesis of leukotrienes from arachidonic acid. By blocking this pathway, zileuton reduces leukotriene production, thereby decreasing inflammation and mitigating the inflammatory response induced by LPS. This explains the significant modulation of arachidonic acid metabolism observed in our data. (2) Sphingolipid metabolism: LPS can alter sphingolipid metabolism, affecting cell membrane integrity, signaling, and the production of sphingolipid-derived mediators involved in inflammation and cell survival [33,34]. The modulation of sphingolipid metabolism by zileuton suggests secondary effects beyond leukotriene inhibition. Changes in this pathway could indicate alterations in cell signaling pathways and membrane dynamics in immune responses. Zileuton may stabilize cell membranes and modulate inflammatory signaling by influencing sphingolipid metabolism, contributing to a broader anti-inflammatory effect. These findings suggest that zileuton not only targets leukotriene production but also has broader implications on cellular metabolism and signaling, enhancing its anti-inflammatory properties.

Cell metabolomics analysis is crucial for elucidating the intricate modulation of biological pathways under various physiological and pathological conditions. This is notably exemplified in allergic rhinitis, where significant modulations in histidine metabolism and arachidonic acid metabolism pathways are observed. By employing targeted drug interventions with triprolidine and zileuton, modulation of these specific pathways has been documented, offering invaluable insights into these therapeutic agents’ molecular mechanisms of action. This study underscores the pivotal role of metabolite regulation in unraveling the complexities of disease pathogenesis and therapeutic interventions, thereby advancing our understanding of the molecular mechanisms underlying physiological responses and treatment outcomes.

This study offers several notable strengths. It employs advanced UHPLC-QTOF-MS technology, enabling highly efficient separation and accurate mass detection, which enhances metabolite identification and semiquantitation. A semiquantitative approach was chosen over absolute quantitation due to the lack of internal standards for all metabolites and variability in response factors across compounds. To address this, a stable isotope-labeled internal standard was used to normalize peak areas and calculate relative quantitation. While this method does not provide absolute quantitation, it allows for reliable comparison of metabolite levels across experimental conditions, aligning well with the study’s objectives. The study successfully identifies significantly regulated metabolites and elucidates key metabolic pathways, providing valuable insights into the underlying metabolic dynamics. Furthermore, it establishes a testing model for potential therapeutic agents, offering insights into their efficacy and molecular mechanisms of action.

However, this study has certain limitations. The single time-point data acquisition strategy captures only momentary snapshots of the dynamic metabolomic landscape, potentially missing critical changes during allergic rhinitis progression or the effects of positive controls. Additionally, using a reverse-phase LC system, while effective for analyzing non-polar to moderately polar metabolites, limits the coverage of the entire metabolome. Despite this, the study successfully identified crucial intermediates in key pathways (e.g., MyD88-dependent, NF-κB, and MAPK pathways) highly relevant to inflammatory and allergic responses. Furthermore, the study focuses solely on metabolite profiling and does not assess protein abundance or activity, which are pivotal in regulating metabolic processes. Integrating proteomic analyses could provide a more comprehensive understanding of the interplay between metabolic and proteomic pathways.

We acknowledge the limitation of using only two biological replicates in this study. This decision was guided by the exploratory nature of the investigation, which aimed to establish an initial metabolomic profile of allergic rhinitis and identify potential metabolic pathways of interest. While increasing replicates would enhance statistical power, we reasoned that using a cell line with a consistent genetic background would minimize biological variability. Additionally, the resource-intensive nature of UHPLC-QTOF-MS-based metabolomics, including costs for consumables, instrument time, and data analysis, necessitated a focused experimental design. To mitigate variability from sample handling and preparation, a rigorous three-step normalization procedure was implemented, including (i) cell counting to ensure consistent cell numbers across replicates, (ii) cell protein content normalization to account for differences in cell growth, and (iii) internal standard calibration to correct instrumental variations during metabolite analysis. Multivariate statistical analyses such as principal component analysis (PCA) (Figure 3) and coefficient of variation (CV) for concentration measurements (Table 3) supported the method’s reproducibility and reliability. These analyses demonstrated excellent consistency across replicates. Furthermore, the observed metabolic changes align with known biochemical pathways in allergic rhinitis, reinforcing the biological relevance of the findings.

Accurate metabolite identification posed a challenge due to sample complexity and platform limitations. Although high-resolution mass spectrometry achieved accurate mass measurements (<2 ppm in MS/MS mode) and utilized spectral libraries for annotation, the absence of analytical standards remains a constraint. Stringent criteria, including isotopic pattern matching and MS/MS fragmentation matching (when available), were applied to enhance reliability. Despite these efforts, some metabolites may remain ambiguously identified or unannotated.

The study did not include drug-treated controls (e.g., cell-treated triprolidine or zileuton without LPS stimulation). While such controls would provide insights into the drugs’ effects on baseline metabolism, the primary objective was to investigate metabolic changes in LPS-induced inflammation and the drugs’ modulatory effects in this context.

While this study provides significant insights into metabolite regulation and pathway modulation, a more holistic approach that includes continuous monitoring and proteomic analysis could further enhance our understanding of the complex biological processes involved.

## 5. Conclusions

This study provides significant insights into the metabolomic landscape of mast cells under conditions of allergic rhinitis and the impact of therapeutic interventions. Utilizing UHPLC-QTOF-MS-based untargeted and targeted metabolomics, we identified 44 significantly regulated metabolites, including histamine, leukotrienes, prostaglandins, thromboxanes, and ceramides. Pathway analysis revealed significant modulations in arachidonic acid metabolism, histidine metabolism, and sphingolipid metabolism, which are critical in the inflammatory response associated with allergic rhinitis.

Our findings demonstrated that LPS-induced stimulation of mast cells results in significant metabolic changes indicative of an inflammatory state. Treatment with triprolidine and zileuton modulated these metabolic pathways, effectively reversing the metabolic shifts induced by LPS. Triprolidine primarily affected histidine and sphingolipid metabolism, whereas zileuton specifically targeted arachidonic acid and sphingolipid metabolism.

Integrating advanced metabolomics techniques in this study provided comprehensive insights into the complex biochemical processes underpinning allergic rhinitis. This research not only enhances our understanding of mast cell metabolism in allergic responses but also highlights the potential of metabolomics in evaluating the efficacy of therapeutic agents. Future studies incorporating continuous monitoring and proteomic analysis could further unravel the dynamic interplay between metabolites and proteins in allergic inflammation.

## Figures and Tables

**Figure 1 biomolecules-15-00109-f001:**
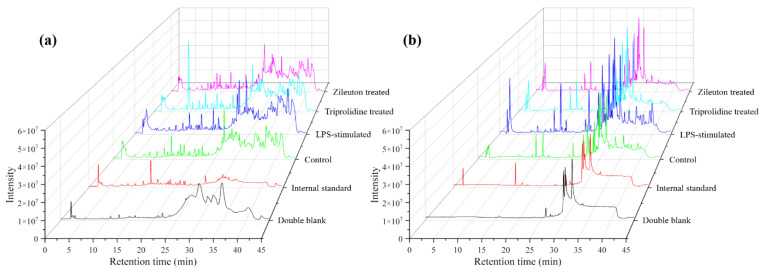
The representative total ion chromatograms (TICs) of the double blank, internal standard solution (2 µg/mL), and the cell samples of the untreated control, LPS-stimulated, triprolidine-treated LPS-stimulated, and zileuton-treated LPS-stimulated groups. (**a**) Positive ionization mode and (**b**) negative ionization mode.

**Figure 2 biomolecules-15-00109-f002:**
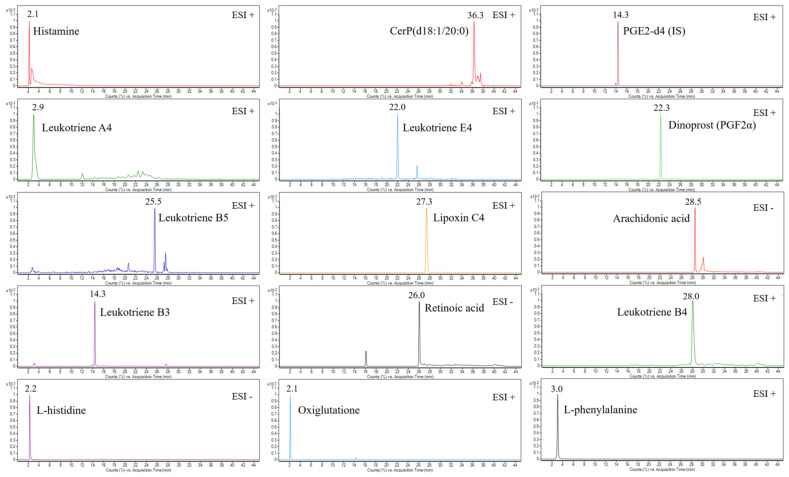
The representative extracted ion chromatograms (EICs) of some identified metabolites in either positive or negative ESI acquisition mode and the internal standard (2 µg/mL) in the LPS-stimulated mast cell samples.

**Figure 3 biomolecules-15-00109-f003:**
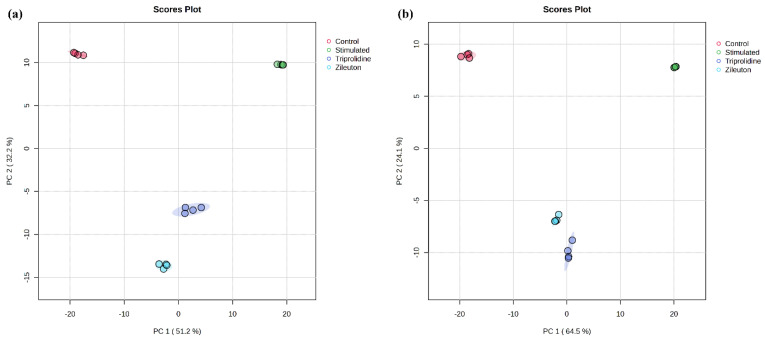
The unsupervised 2D PCA plot of multivariate data analysis on all molecular features detected by UHPLC-QTOF-MS-based untargeted metabolomics in mast cells under various experimental conditions and data acquisition modes: (**a**) positive and (**b**) negative.

**Figure 4 biomolecules-15-00109-f004:**
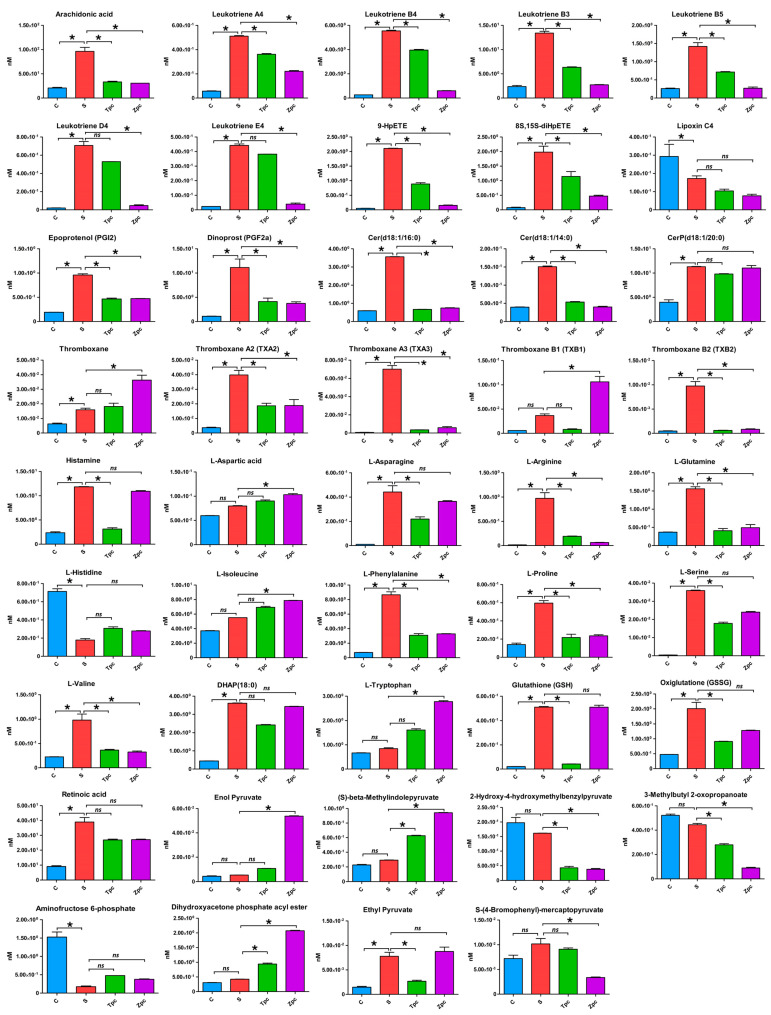
Comparison of metabolic changes of 44 regulated metabolites in mast cells under various experimental conditions: untreated or control (C) vs. LPS-stimulated (S), LPS-stimulated vs. triprolidine-treated post-LPS stimulation (Tpc), and LPS-stimulated vs. zileuton-treated post-LPS stimulation (Zpc). (*: *p*-value < 0.05 and log2 fold change > 2; ns: statistically not significant with *p*-value ≥ 0.05).

**Figure 5 biomolecules-15-00109-f005:**
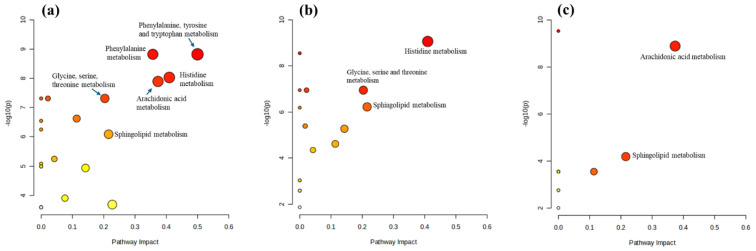
The pathway analysis of various paired experimental conditions. (**a**) Control vs. LPS-stimulated, (**b**) LPS-stimulated vs. triprolidine-treated, and (**c**) LPS-stimulated vs. zileuton-treated. (Red color denotes high −log10(p) and yellow color denotes low −log10(p) values. Bigger circle denotes high pathway impact score while smaller circle denotes lower pathway impact circle).

**Figure 6 biomolecules-15-00109-f006:**
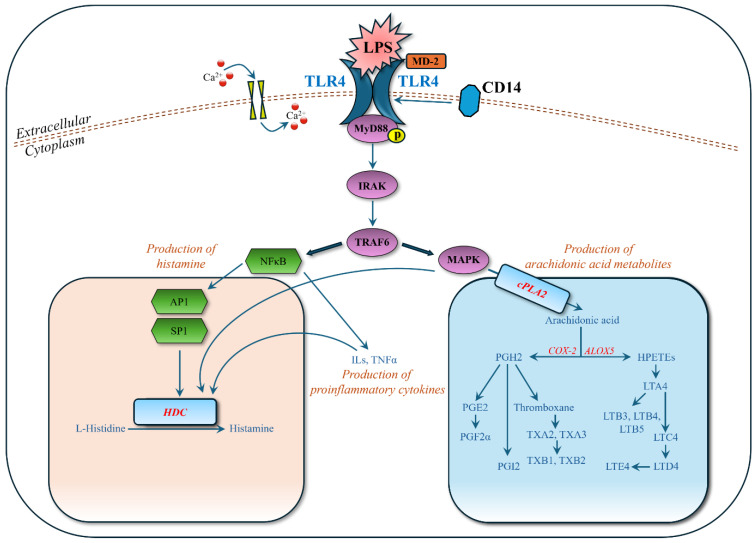
Activation of MyD88 (myeloid differentiation)-dependent pathways by binding LPS (lipopolysaccharide) to TLR4 (Toll-like receptor 4). Other abbreviations: Ca^2+^: calcium ion; IRAK: interleukin-1 receptor-associated kinase; TRAF6: TNF receptor-associated factor 6; MAPK: mitogen-activated protein kinase; NFκB: nuclear factor κB; AP1: activator protein 1; SP1: specificity protein 1; HDC: histidine decarboxylase; cPLA2: cyclic phospholipase A2; COX-2: cyclooxygenase-2; ALOX5: arachidonate 5-lipoxygenase; ILs: interleukins; TNFα: tumor necrosis factor α; MD-2: myeloid differentiation factor 2; CD14: cluster of differentiation 14.

**Table 1 biomolecules-15-00109-t001:** The matrix effects of mast cell samples under various experimental conditions on mass spectrometric detection.

Experimental Condition	ESI Mode	Matrix Factor ± SD (*n* = 4)
Control	+	0.98 ± 0.02
	−	1.12 ± 0.00
LPS-stimulated	+	0.82 ± 0.04
	−	1.05 ± 0.02
Triprolidine positive control-treated	+	0.92 ± 0.02
	−	1.00 ± 0.01
Zileuton positive control-treated	+	0.97 ± 0.02
	−	1.01 ± 0.00

## Data Availability

The data used for the generation of this manuscript are contained within the article. The corresponding author will provide any required raw data upon request.

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
