# Peer review of "Comprehensive Metabolomics in Mouse Mast Cell Model of Allergic Rhinitis for Profiling, Modulation, Semiquantitative Analysis, and Pathway Analysis"

_biomolecules, 2025, doi:10.3390/biom15010109_

Round 1
Reviewer 1 Report
Comments and Suggestions for Authors
The study by Patil and Su investigates changes in cellular metabolism in a cell culture model of allergic rhinitis. Murine mast cells were stimulated with lipopolysaccharide (LPS) and changes in metabolite levels were detected by untargeted liquid chromatography-high resolution mass spectrometry (LC-QqTOF-MS). LPS-stimulated cells were also treated with two antiallergic drugs, and changes in metabolite levels were compared to untreated LPS-stimulated cells. LC-MS data for these four treatments was analysed statistically and a pathway analysis tool was employed to identify effects on a biochemical pathway level.
The study is generally well designed and presented. It uses high-quality methods and tools for metabolomic investigation. Data analysis including metabolite identification and multivariate statistics follows common standards. Only two biological replicates were used, but this limitation appears acceptable within the scope of the study.
The paper is well written and I have only few minor comments.
l. 129: Frozen cells were first resuspended in water for metabolite extraction, and the usual organic solvent (acetonitrile) was only added afterwards. Did the authors check that this did not result in metabolite degradation? Shock freezing cells might eliminate a large percentage of enzymatic activity but some activity might remain. Did the authors investigate this?
Table 3 could be combined with Table 4. Consider showing log-fold changes instead of molar concentration, and moving the latter to Table S1. Also consider adding trend indicators, such as upwards arrow for increased metabolite levels. Also indicate clearly that molar concentrations are *rough* estimates as ES ionization efficiencies vary widely for small molecules.
ll. 623-636: The authors should include the following aspects in their discussion of the limitations of the study.
a) use of only two biological replicates
b) metabolite coverage: polar metabolites only partly covered by reverse phase LC
c) quantification approach gives only rough estimates due to varying response factors
d) limitations of metabolite
e) study does not include drug-treated control cells (w/o LPS stimulation)
Reviewer 2 Report
Comments and Suggestions for Authors
Allergic rhinitis is a disease with a high incidence rate and a serious impact on the quality of life of patients. The study of the mechanism of allergic rhinitis and the characteristics of anti-allergic drugs is of great significance for the discovery of new drugs for the treatment of allergic rhinitis. This study used UHPLC-QTOF/MS untargeted and targeted metabolomics techniques to investigate the metabolic changes of mouse mast cells (MC/9) under LPS induced allergic rhinitis conditions. The results showed that most of the 44 significantly altered metabolites could be inhibited by anti-allergic drugs Triprolidine and Zileuton. This has certain reference value for our understanding of the mechanism of mast cells involved in allergic reactions and the establishment of cell models for evaluating anti allergic drugs. But there are some questions in the manuscript that need to be answered by the author.
1. There are many types of inflammatory cells involved in allergic rhinitis, including mast cells, eosinophils, basophils, and epithelial cells. What are the considerations for choosing mast cells as a representative chronic rhinitis cell model? Have you considered the role of other cells (such as epithelial cells) in allergic rhinitis?
2. What are the considerations for selecting mouse mast cells (MC/9) as a cell model? Why not choose HMC-1 cell line and LAD2 cell line? This may be closer to human diseases.
3. Please provide a general description of the advantages and disadvantages of using UHPLC-QTOF/MS technology. What are the advantages of this method, especially when using mast cells as a cell model to study the efficacy of drugs?
4. The action stages of two drugs (Triprolidine and Zileuton) should be reflected in the schematic diagram (Figure 6).
Reviewer 3 Report
Comments and Suggestions for Authors
The presented paper describes a metabolomics study conducted in a mouse model of allergic rhinitis, focusing on mast cells. UHPLC-QTOF/MS-based untargeted and targeted metabolomics approaches were employed to comprehensively profile the metabolic changes associated with the induction and modulation of the allergic response in the mouse mast cell model in response to two different treatment conditions. A total of 44 significantly regulated metabolites, including histamine, leukotrienes, prostaglandins, thromboxanes, and ceramides, were identified along with their associated biochemical pathways, providing insights into the underlying mechanisms of allergic rhinitis development and progression. Overall, the manuscript is neat, and the presentation is clear, however, some additional experiments and more rigorous data analysis could further strengthen the conclusions.
- My major concern is regarding the biological replicate, the authors used two biological replicates in each condition which may not be sufficient to draw definitive conclusions, especially for the semiquantitative analysis and pathway enrichment. I would suggest including more biological replicates to improve the statistical robustness of the results.
- Additionally, Why the authors only employ PCA as the multivariate analysis approach? I would also suggest performing other multivariate techniques such as OPLS-DA or random forest to further validate the differences between the experimental conditions.
- Regarding the identification of the metabolites, the authors should provide more details on the identification confidence level for each metabolite as per the metabolomics standards initiative guidelines. They mention they only search the molecular features against METLIN and lipid databases (Line 223) but what about tandem MS/MS fragmentation data?
Overall, the manuscript is a nice piece of work but requires additional data and more robust statistical analysis to strengthen the conclusions.
Round 2
Reviewer 3 Report
Comments and Suggestions for Authors
Accept